# Critical Overview of Hepatic Factors That Link Non-Alcoholic Fatty Liver Disease and Acute Kidney Injury: Physiology and Therapeutic Implications

**DOI:** 10.3390/ijms232012464

**Published:** 2022-10-18

**Authors:** Le Chen, Xiaodong Lv, Min Kan, Ruonan Wang, Hua Wang, Hongmei Zang

**Affiliations:** 1Inflammation and Immune Mediated Diseases Laboratory of Anhui Province, Anhui Institute of Innovative Drugs, The Key Laboratory of Anti-Inflammatory of Immune Medicines, School of Pharmacy, Anhui Medical University, Hefei 230032, China; 2Department of Oncology, The First Affiliated Hospital of Anhui Medical University, Hefei 230032, China

**Keywords:** nonalcoholic fatty liver, acute kidney injury, renal function, hepatic factor, hepatic-renal axis

## Abstract

Non-alcoholic fatty liver disease (NAFLD) is defined as a combination of a group of progressive diseases, presenting different structural features of the liver at different stages of the disease. According to epidemiological surveys, as living standards improve, the global prevalence of NAFLD increases. Acute kidney injury (AKI) is a class of clinical conditions characterized by a rapid decline in kidney function. NAFLD and AKI, as major public health diseases with high prevalence and mortality, respectively, worldwide, place a heavy burden on societal healthcare systems. Clinical observations of patients with NAFLD with AKI suggest a possible association between the two diseases. However, little is known about the pathogenic mechanisms linking NAFLD and AKI, and the combination of the diseases is poorly treated. Previous studies have revealed that liver-derived factors are transported to distal organs via circulation, such as the kidney, where they elicit specific effects. Of note, while NAFLD affects the expression of many hepatic factors, studies on the mechanisms whereby NAFLD mediates the generation of hepatic factors that lead to AKI are lacking. Considering the unique positioning of hepatic factors in coordinating systemic energy metabolism and maintaining energy homeostasis, we hypothesize that the effects of NAFLD are not only limited to the structural and functional changes in the liver but may also involve the entire body via the hepatic factors, e.g., playing an important role in the development of AKI. This raises the question of whether analogs of beneficial hepatic factors or inhibitors of detrimental hepatic factors could be used as a treatment for NAFLD-mediated and hepatic factor-driven AKI or other metabolic disorders. Accordingly, in this review, we describe the systemic effects of several types of hepatic factors, with a particular focus on the possible link between hepatic factors whose expression is altered under NAFLD and AKI. We also summarize the role of some key hepatic factors in metabolic control mechanisms and discuss their possible use as a preventive treatment for the progression of metabolic diseases.

## 1. Introduction

Together, the liver and kidney are extremely important for the maintenance of systemic homeostasis in the organism. These organs interact under both physiological and pathological conditions, and it is widely accepted that impairment of the function of one organ exerts pathological effects on the other [1]. A variety of factors lead to renal dysfunction in the presence of liver disease. The causal factors have various origins, such as hepatitis B and C virus infection, and chronic excessive alcohol intake [2,3,4]. In the last decade, many studies have demonstrated the association between liver disease and the development of nephropathies, such as a significantly increased risk of chronic kidney disease (CKD) in patients with non-alcoholic fatty livery disease (NAFLD), between 20% and 50%, compared to that in patients without NAFLD, between 5% and 25% [5,6]. A recent meta-analysis showed that in patients with NAFLD, elevated liver stiffness was associated with increased odds of renal outcomes [7]. Widespread use of VCTE to screen for advanced fibrosis may help identify patients at risk for end-stage renal disease [8]. Common triggers of CKD include nephrotoxic drug (cisplatin, gentamicin, adriamycin, etc.) use and acute kidney injury (AKI), which ultimately lead to the deterioration of renal function and destruction of renal structures. These outcomes are associated with an irreversible loss of renal cells and units at the onset of CKD [9].

The liver is an important endocrine organ that secretes hepatic factors. Its unique structure and extensive blood perfusion allow it to send molecular signals to, and thus to communicate with, distant tissues or organs via the blood flow. Dysregulation of the pathway of inter-organ communication centered on the liver may lead to improper regulation of energy homeostasis and, eventually, to metabolic dysfunction-like diseases, which provides some theoretical support to explain the high incidence of CKD and AKI under NAFLD. Therefore, deciphering the mechanisms whereby hepatic factors are expressed and interact with distal tissues or organs is crucial for the prevention and treatment of metabolic diseases. In the current review, we briefly characterize NAFLD and AKI and crosstalk between them, focusing on hepatic factors affected by NAFLD that may link NAFLD with AKI. We also highlight the prospective use of hepatic factors as possible targets for developing preventive treatment for the progression of metabolic diseases.

## 2. NAFLD

NAFLD presents with major lipid accumulation and steatosis of the liver and encompasses nonalcoholic fatty liver, nonalcoholic steatohepatitis (NASH), and cirrhosis [10]. The incidence of NAFLD, a growing global public health problem, with a current global prevalence among adults of approximately 25%, is expected to increase further in the next decade posing a serious economic challenge to the affected individual’s family and society [11,12]. The main difference between NAFLD and another common liver disease, alcoholic liver disease (ALD), is the involvement of excess alcohol in the development of hepatic steatosis in the latter, with more than 5% of hepatocytes affected. The pathological course of ALD is similar to that of NAFLD, and includes simple alcoholic steatosis, alcoholic hepatitis, fibrosis, and cirrhosis. Fortunately, steatosis caused by excessive alcohol consumption can be reversed approximately 1 month of abstinence from alcohol [13]. Without medical intervention, NAFLD and ALD both lead to hepatocellular carcinoma (HCC), a cancer with extremely short patient survival upon diagnosis, with little to no treatment, and one of the three leading causes of cancer-related deaths worldwide [14].

The pathogenesis of NAFLD stems from a wide range of sources. Traditionally, high food intake, poor dietary habits, metabolic dysfunction, oxidative stress, and genetic factors have been viewed as pathophysiological mechanisms of NAFLD. In recent years, studies focusing on the intestinal flora revealed that, surprisingly, alterations along the gut–liver axis may also be an important factor in the development of NAFLD [15]. Fibrosis repair and immune system-driven inflammation play a key role in the further development of NAFLD. Fibrosis development is accompanied by substantial damage to the liver structure and function. During NASH with fibrosis, a large number of functional cells are transformed into non-functional fibroblasts, with irreversible damage to liver function, which the patients should be aware of. Currently, a non-invasive scoring system is being established for the detection of fibrosis indicators in the blood and their combination, as a means to assess the degree of liver fibrosis, but the heavy dependence on blood sampling is not conducive to repeated multiple examinations. Transient elastography has been developed as an alternative. It is accepted by many patients because of its non-invasiveness and reproducibility, and by healthcare professionals because it facilitates patients with underlying liver fibrosis and staging the disease in patients who develop fibrosis [16].

Inflammation and insulin resistance (IR) play an important role in NAFLD. The effects of NAFLD-mediated metabolic syndrome (MetS) characterized by cytokine storm and IR are not limited to the liver. However, under pathological conditions, the circulating hepatic factors potentiate the systemic impact of NAFLD [17]. Hepatic lipid accumulation and steatosis, as the initiating stage of the NAFLD disease course, are important for controlling disease progression. According to recent studies, IR plays a central role in hepatic steatosis, and calls to rename NAFLD as metabolic-associated fatty liver disease as a way to emphasize the close association between obesity, type 2 diabetes mellitus (T2DM), and the MetS, have intensified in the last two years [18]. However, a cross-sectional study based on US adults showed a prevalence of 37.1% and 39.1% for NAFLD and MAFLD, respectively. This suggests a high level of agreement between these two definitions [19]. The link is an important step toward the understanding of NAFLD, and we look forward to more evidence to redefine NAFLD. 

## 3. AKI

AKI is an acute and rapidly injurious disease. It leads to a sudden decline and loss of kidney function, as evidenced by a sudden decrease in the glomerular filtration rate (GFR), an increase in serum creatinine (Scr) levels, and a disruption of body homeostasis [9]. AKI seriously affects survival. Approximately 13 million individuals are diagnosed annually worldwide, and AKI has become a major medical problem that urgently needs to be addressed.

AKI can be triggered by a variety of factors, such as infection, hypovolemic shock, sepsis, nephrotoxic drugs, or surgery, and the disease outcome is primarily related to the severity and duration of the trigger [20]. Despite the widespread effects of AKI, effective treatment options are lacking. In addition to the ensuing rapid renal failure, another important reason why AKI is of great concern is its frequent occurrence in combination with other serious diseases, which can be devastating for the affected individual. Unfortunately, the mechanisms underlying multiple AKI in critically ill patients are still unclear and effective means of suppression are lacking, making the prevention and early detection of AKI in critically ill patients particularly important. No uniform criteria have been established for the adjudication of renal failure, and more than 35 different definitions of acute renal failure are used in the available literature [21]. The currently commonly used functional criteria are a 50% increase in Scr, a 0.3 mg/dL (26.5 mol/L) increase in Scr over 2 d, or oliguria for 6 h, etc., lasting 7 d, but there is no uniform standard for renal structural changes in AKI [9].

Because of the unique timing of AKI, early detection is crucial. However, the lack of a single biomarker recognized as an early marker of renal impairment contributes to the difficulties in the early diagnosis of AKI. Based on the traditional use of KIM-1 and matrix metalloproteinase tissue inhibitory factor (TIMP)-1 as detectors of kidney injury, some new indicators, such as insulin-like growth factor-binding protein (IGFBP) 7, are continuously discovered to facilitate early identification of AKI [22]. Some investigators have achieved promising results using plasma-soluble urokinase fibrinogen activator receptor and neutrophil gelatinase-associated lipid transport protein (NGAL) [23,24]. Nonetheless, in spite of some progress, the currently used widespread detection of kidney injury based on urine volume and Scr often lags behind changes in the actual renal function, resulting in less-than-optimal results. In general, the development of highly specific and sensitive detection markers presents a serious challenge.

## 4. The Liver and Kidney Cross-Talk

Organ cross-talk is a relatively novel concept describing organ-to-organ and organ-to-tissue signaling. Dysregulation of the cross-talk can disrupt systemic homeostasis, which may induce disease (Figure 1). The liver and kidneys, two important organs, have been found to engage in such cross-talk. For example, in hepatorenal syndrome, the progressive course of liver disease always leads to renal failure. As another example, activation of the renin-angiotensin-aldosterone system and sympathetic nervous system in patients with liver disease as a compensatory mechanism for visceral vasodilation may trigger AKI caused by renal vasoconstriction and inadequate perfusion. Further, fatty liver causes the translocation of the intestinal flora, leading to a systemic inflammatory response along the hepatic-intestinal-renal axis, which is also a key mechanism inducing renal vasoconstriction and glomerular filtration rate. Conversely, the occurrence of AKI can promote a systemic inflammatory response and oxidative stress, leading to liver damage. For example, elevated serum levels of alanine aminotransferase, asparagine aminotransferase, interleukin, and tumor necrosis factor-α are caused by ischemic injury to the kidney [25]. The above observations reflect a possible interaction between the liver and kidney, suggesting that the mechanism of liver-kidney crosstalk should be elucidated not only under normal physiological conditions, but also under abnormal pathological conditions, to prevent and control the ensuing potential metabolic diseases.

## 5. Hepatic Factors

The liver, as the main secretory organ, has long been the focus of extensive research. It releases organ-specific cytokines, also known as hepatic factors or hepatokines, that have autocrine, paracrine, and endocrine functions (Table 1). These molecules significantly impact metabolic homeostasis but are also a major source of new biomarkers for the diagnosis of metabolic diseases [26]. Previous studies have focused on the role of hepatic factors in coagulation and anticoagulation, but recent studies have reported a link between hepatic factors and metabolic diseases. Interestingly, NAFLD affect the secretion of liver factors [27].

Hepatic factors play an irreplaceable role in coordinating systemic energy metabolism, and the liver adjusts secretion in response to changes in the body, as evidenced, e.g., by a differential expression of hepatic factors in patients with T2DM and NAFLD compared with that in healthy individuals. Because of its powerful secretory function, the liver plays an important role in maintaining systemic homeostasis, and the dysregulation of hepatic secretion may lead to improper regulation of energy homeostasis and, eventually, to metabolic dysfunction. Unfortunately, little is known about the regulatory mechanisms of hepatic factor involvement in the MetS. Hence, revealing the liver-centered inter-organ interaction mechanisms is a formidable research challenge for the future. Below, we summarize the mechanisms that link liver and kidney to determine predisposition to AKI in patients with NAFLD.

### 5.1. Angiopoietin-Like Proteins (ANGPTLs)

ANGPTLs (ANGPTL1 to ANGPTL8) are circulating hepatic factors. Their structure is similar to that of members of the angiopoietin family of proteins [28]. ANGPTLs are expressed in many tissues, including the liver and vascular system, with the related studies mainly focusing on inflammatory disease control, lipid regulation in metabolic diseases, and angiogenesis [29].

ANGPTL1, a vasopressor, was the first protein from the ANGPTL family to be identified. Initially, it was overwhelmingly studied in the context of the cardiovascular system, because of its expression mainly in vascularized tissues [28]. ANGPTL1 was later found to be downregulated in a variety of cancers [30]. As has been recently shown, ANGPTL1 present in exosomes inhibits liver metastasis of colorectal cancer by regulating the secretion of hepatic blast cells [31].

ANGPTL2 has been detected in the adipose tissue, kidney, and skeletal muscle. Fibrosis in tissues, such as the liver and kidney, is closely related to chronic inflammation, and ANGPTL2 may play an important role in the fibrotic response of organs. Additionally, ANGPTL2 expression inhibits the secretion of inflammatory cytokines by renal tissues, which in turn inhibits renal autophagy and regulates renal fibroblast activation [32].

ANGPTL3 is a specific secretory factor expressed and secreted mainly by hepatocytes, although is expressed both in the liver and kidney. It is a polypeptide of 460 amino acids with a characteristic angiopoietin structure [33]. ANGPTL3 acts as a regulator of lipid metabolism inhibits lipoprotein lipase and has become a promising target for the regulation of lipid levels [34]. The lipoprotein lipase family consists mainly of pancreatic lipase (PL), lipoprotein lipase (LPL), and hepatic lipase (HL). LPL is involved in the processing of triglyceride-rich lipoprotein (transport and metabolism), for energy production and lipid storage [35]. ANGPTL3 acts by itself or in a protein functional complex with ANGPTL8 to inhibit LPL activity and regulate lipid metabolism.

ANGPTL4 is mainly secreted into the circulation by white and brown adipose tissue and liver tissue [36]. Similar to ANGPTL3, ANGPTL4 plays an important role in the regulation of lipid storage and catabolism. Induction of ANGPTL4 in skeletal muscle can promote plasma triglyceride as an energy source to provide energy for active muscle tissue by down-regulating AMP-activated protein kinase-mediated mechanism [37].

ANGPTL5 is mainly expressed in adipose tissue. According to published studies, it can regulate triglyceride metabolism to alleviate the MetS caused by energy imbalance, a finding further supported by the increase in triglyceride levels in obesity and diabetes. In a recent study on exercise and circulating ANGPTL5 levels, the relationship between ANGPTL5 and prevalent cardiovascular disease risk factors was investigated by comparing circulating ANGPTL5 levels in obese normal-weight adolescents. Elevated levels of ANGPTL5 were detected in obese adolescents and were associated with cardiovascular disease risk factors, high-sensitivity C-reactive protein, and oxidized low-density lipoprotein [38].

ANGPTL6 is secreted into the circulation mainly by the liver. It may play an important role in human metabolism as such, as it is important for the maintenance of glucose homeostasis. It has been reported that targeted activation of ANGPTL6 leads to increased insulin sensitivity, increased energy expenditure, and prevention of diseases related to IR, e.g., a reduced risk of developing diabetes [39]. In another study, a positive association between ANGPTL6 levels in the serum and fasting glucose levels in patients with T2DM was detected [40]. In addition, high levels of circulating ANGPTL6 in patients with HCC are significantly associated with poorer disease treatment outcomes, when compared with other disease indicators, suggesting ANGPTL6 as a novel diagnostic and prognostic marker for HCC [41]. By applying statistical analysis to prior clinical observations, ANGPTL6 can be used as both an independent risk factor for the induction of HCC and a prognostic indicator for treatment, with the effectiveness of treatment verified by measuring the ANGPTL6 expression in patients after the treatment. Overall, one of the major advantages of ANGPTL6 compared to the other indicators is its superior sensitivity.

ANGPTL8 is mainly secreted by the liver and often acts as a physiological inhibitor of LPL, a key enzyme of plasma triglyceride metabolism. Hence, ANGPTL8 plays a key role in lipoprotein and triglyceride metabolism [42]. Of note, ANGPTL8 inhibits LPL only when combined with ANGPTL3, with no effect of ANGPTL8 alone [43]. Further, according to recent studies, increased serum levels of ANGPTL8 correspond to the severity of NAFLD [44].

### 5.2. Fetal Globulin

Fetuin-A and fetuin-B are glycoproteins that share 22% sequence similarity and belong to the same cystatin superfamily of proteins. Fetuin-A is released mainly by the liver, and the first liver-derived protein that has been shown to be associated with metabolic diseases [21]. It is common for a protein to play opposite roles under different environmental conditions. Accordingly, fetuin-A plays multiple positive roles in health by regulating the secretion of some inflammatory cytokines and exosomes. However, it plays negative roles in the course of some diseases, such as obesity, diabetes, and fatty liver [45]. In individuals with elevated levels of liver fat, circulating levels of fetuin-A are also increased, compared to those in individuals with normal liver fat levels. According to some studies, individuals with elevated circulating levels of fetuin-A are more likely to develop T2DM than the general population [46,47]. Further, an increased secretion of fetuin-A in adipose tissue has been noted in T2DM patients, which was found to bind to toll-like receptors, ultimately leading to an inflammatory response and IR in T2DM patients [48]. These observations have been confirmed on a molecular level. Namely, a significant increase in the expression of fetuin-A transcript levels was noted in hepatocytes of patients with hepatic steatosis compared to healthy controls, corresponding to significantly elevated levels of fetuin-A in the serum [49]. Fetuin-A may also be involved in the pathogenesis of CKD, as it is abundant in the serum, and CKD is often associated with hyperphosphatemia, linking fetuin-A to vascular calcification [50]. All these studies demonstrate a close association of fetuin-A with vascular calcification. Calcification is more common and serious among CKD patients than in the general population. A progressive increase in coronary artery calcification leads to impaired blood circulation, resulting in inadequate blood perfusion of the kidneys and leading to a progressive loss of kidney function [51]. The imbalance between predisposing factors, such as hypercalcemia, and inhibiting factors, such as fetuin-A, is critical to the development of vascular calcification [52]. These pieces of evidence suggest that vascular calcification may be one of the mechanisms that induce renal injury in patients with liver disease.

Fetuin-B expression is increased in the liver of patients on a high-fat diet. The expression of both fetuin-B and fetuin-A is altered in the development of fatty liver compared to normal subjects. However, unlike fetuin-A, fetuin-B is not involved in pro-inflammatory signaling and macrophage activation induced by NAFLD. Instead, it may potentiate steatosis and induce the development of IR. According to a recent study, silencing of fetuin-B gene improves glucose tolerance in obese mice without altering the body weight, suggesting the specificity of this regulation.

There are two pathways to increasing insulin levels: increasing the amount of insulin produced and increasing the efficiency of glucose utilization. Instead of acting via the classical mechanism of insulin dependence to regulate blood glucose levels, fetuin-B may act via another insulin-independent mechanism, e.g., involving efficient glucose utilization [53]. The exact mechanism remains to be investigated. Clinical observations suggest a potential link between high levels of circulating fetuin-B and the development of diabetes and coronary artery disease. Indeed, the expression of fetuin-B is upregulated under pathological conditions and then exacerbates myocardial burden by inhibiting insulin signaling in diabetic mice [54]. The above studies suggest that the effects of fetuin secreted in liver disease are not limited to the liver but can exert extrahepatic effects via circulation. The most important extrahepatic outcomes are renal and cardiovascular diseases represented by diabetic nephropathy and cardiovascular disease, respectively.

Despite their similarity, fetuin-A and fetuin-B play different roles in the pathophysiology of fatty liver. The expression of fetuin-A and fetuin-B is both upregulated in the presence of hepatic steatosis in human, but their regulation of glucose homeostasis is different fetuin-A alters insulin signaling, while the fetuin-B mechanism may be related to glucose utilization [55]. Considering that fetuin-A and fetuin-B impact lipid accumulation and lipodegradation, and their altered expression in NAFLD, fetuin-A and fetuin-B are possibly involved in the regulation of renal and cardiovascular function in NAFLD.

### 5.3. Fibroblast Growth Factor (FGF)

FGF are a class of proteins secreted primarily by the liver and involved in multiple biological functions in vivo [56]. They play a role in the regulation of developmental processes, including brain and limb development [57]. Further, there is substantial evidence that while FGF signaling plays an important role in the maintenance of normal liver function, it is dysregulated in patients who develop liver disease [55].

FGF1 has been used in recent studies as one of the potential safe candidate targets for restoring normoglycemia in T2DM [58]. Pronounced hyperglycemia and IR were noted in *FGF1* knockout mice that consumed high-fat foods, suggesting that FGF1 may play a regulatory role in nutritional homeostasis [59].

Elevated FGF2 expression was detected in the tissues of obese patients, and significantly elevated protein levels in the serum of patients with cirrhosis and HCC [60].

FGF19 is involved in the regulation of lipid metabolism and glucose homeostasis [61]. A significantly increased expression of FGF19 was observed in tissue samples from patients with fatty liver compared to liver tissue samples from healthy subjects. In a recent preclinical study, the FGF19 analog Aldafermin was shown to reduce liver fat and exert beneficial effect on fibrosis in a trial involving patients with NASH [62].

FGF21 is produced by the liver. It enters the bloodstream and is an important regulatory molecule controlling hepatic lipid metabolism and glucose metabolism [63]. Elevated FGF21 expression was observed in the serum of patients with NAFLD, and both serum levels of FGF21 protein and liver levels of FGF21 transcripts were positively correlated with triglyceride levels in the liver [64]. Currently, renal elimination is the main pathway for maintaining FGF21 serum levels, and the finding that FGF21 serum concentrations are elevated in patients with renal failure validates this conclusion [65]. In addition, activation of JNK signaling in adipocytes regulates circulating levels of FGF21 in vivo, which could be a potential compensatory mechanism in the event of impaired organ interaction [66]. Initially, FGF19 and FGF21 were defined as new energy-regulating endocrine messengers, in the context of ameliorating hyperglycemic symptoms in patients with T2DM. Surprisingly, trials involving human participants revealed their powerful modulatory effects on lipids, and clinical studies have provided evidence for shifting their use to the treatment of NASH and hypertriglyceridemia [67].

### 5.4. Heparin

Hepcidin (HPS) is a hepatocyte-derived fibrinogen-associated protein, and a novel hepatic factor that causes accumulation and degeneration of hepatic fat, and ultimately IR. HPS is a specific mitogenic activator of hepatocytes, and its overexpression in HCC cells inhibits cell growth [68]. HPS is also involved in the development of NAFLD, and elevated HPS expression in patients with NAFLD exacerbates the accumulation of hepatic lipids, thus aggravating the progression of NAFLD [69]. An investigation involving 371 normal, overweight, or obese subjects revealed that HPS expression in overweight or obese subjects is significantly different from that in normal subjects [70]. A recent study of the relationship between circulating HPS levels and liver and kidney function in patients with stable angina pectoris demonstrated an association between increased HPS expression and liver and kidney disease, confirming the involvement of increased plasma HPS levels in the development of diseases, such as NAFLD and CKD, via the peripheral circulation pathway [71].

### 5.5. Retinol-Binding Protein 4 (RBP4)

RBP4 is derived from adipose tissue, transports retinol (vitamin A) into the serum, and may act as a mediator between obesity and IR [72]. Differential expression of RBP4 may contribute to IR and hepatic steatosis by affecting the balance between adipogenesis and consumption in the liver tissue [73]. Changes in RBP4 levels also appear to be associated with the hepatic manifestations of NAFLD and MetS and may also be a major cause of liver injury [74]. A recent study in a genetic mouse model revealed that in mice, circulating RBP4 is produced only by hepatocytes [75]. Further, hepatic secretion of RBP4 does not disrupt glucose homeostasis, indicating that the modest increase in circulating levels of RBP4 in mice observed in obesity and IR might not be the cause of impaired glucose homeostasis. According to observational studies with clinical samples, circulating levels of RBP4 in the CKD population are higher than those in the normal adult population [76]. Consequently, we speculate that the circulating levels of RBP4 may reflect the stage of CKD.

### 5.6. Insulin-Like Growth Factor (IGF) and IGFBPs

As stated in the definition, IGF is structurally and functionally very similar to insulin, and there is evidence that altered IGF signaling may be potentially relevant to HCC progression [77]. IGF-1 is expressed in many tissues, helps to regulate hepatic glucose and lipid metabolism, and reduced expression may contribute to the development of NAFLD [78]. Since most circulating IGF-1 is of hepatic origin, IGF-1 is also considered to be a hepatic factor.

IGFBPs regulate circulating IGF levels by binding to IGF, thereby keeping it in circulation for transport to peripheral tissues. IGFBPs also play an IGF-independent role. For example, IGFBP7 inhibits the expression of proteins associated with the activation of IGF signaling by binding to IGF-1 receptor and acts as a tumor suppressor, and IGFBP7 deficiency may promote HCC [77]. Further, the regulation of IGF by IGF-1 and IGFBPs may also be involved in glycemic control and NAFLD development, and different IGFBPs may have different regulatory effects [78].

IGFBP7 is strongly associated with liver disease. Of note, according to several studies, it is abundantly expressed in the kidney, with differential expression of IGFBP7 observed in the urine of patients with AKI. In addition, we found that IGFBP7 has unique advantages as an early indicator of AKI, and its development, as early detection of AKI has a great potential for application translation. However, the appropriate IGFBP7 threshold has not yet been determined, resulting in a lack of universal acceptance of the definition of AKI in terms of IGFBP7 levels. Meanwhile, while plasma IGFBP7 concentrations can be used to predict renal and cardiac events in participants with T2DM and high cardiovascular risk, more research is needed on the association between circulating IGFBP7 and the progression of diabetic nephropathy [79,80]. While we speculate on the possibility of IGFBP7 acting as a messenger for the liver–kidney inter-organ connection based on the above, the primary goal is to determine the origin of IGFBP7 that acts on the kidney.

### 5.7. Selenoprotein P (SeP)

SeP is a secreted protein of hepatic origin that distributes selenium to the rest of the body via circulation to maintain homeostasis [81]. The main selenoprotein is glutathione peroxidase, which contributes to the control of free radicals at inflammatory sites [82]. SeP is activated the development of NAFLD and exacerbates the development of NAFLD [83]. It is closely related to glucose metabolism. High concentrations of glucose stimulate the pancreas to secrete insulin to regulate blood glucose and stimulate the liver to release SeP to intensify insulin secretion, causing the body to reject insulin [84]. Knockdown of the SeP gene expression in the liver in mice results in an increased insulin sensitivity of the liver [85]. Further, in a well-defined cohort study of non-diabetic subjects with NAFLD, plasma SeP levels were shown to be associated with metabolic disorders and severe liver disease in NAFLD patients without diabetes [84]. These studies suggest that SeP, as a protein secreted by the liver, may play an important role in energy metabolism in humans.

**Table 1 ijms-23-12464-t001:** Hepatic factors, and their role in metabolism and disease.

Hepatokin		Target Organs	Biological Function	References
ANGPTLs	ANGPTL1	Liver, vascular system	Anti-angiogenic, permeability, anti-apoptotic, cancer development	[25,27,28]
	ANGPTL2	Liver, kidney, heart, adipose tissue, skeletal muscle	Angiogenesis, physiological tissue remodeling	[25,29]
	ANGPTL3	WAT, muscle, liver, kidney	Angiogenesis, lipid metabolism	[31,32]
	ANGPTL4	WAT, vascular endothelial cells, liver, kidney,	Lipid metabolism, glucose metabolism,	[25,33,34]
	ANGPTL5	Adipose tissue	Lipid and triglyceride metabolism	[25,35]
	ANGPTL6	Skeletal muscle, WAT, liver	Lipid metabolism, glucose metabolism	[36,38]
	ANGPTL8	Hepatocytes, adipocytes	Lipid metabolism	[39,41]
Fetal globulin	Fetuin-A	Liver, WAT, skeletal muscle, monocytes	Adipose tissue inflammation, IR, vascular calcification	[42,43,44,45,46,47,48,49]
	Fetuin-B	Hepatocytes, myotubes	Steatosis and induction of IR	[50,51]
FGFs	FGF1	Liver, adipose tissue	Regulation of glucose homeostasis	[55]
	FGF19	Liver, adipose tissue	Regulation of bile acid, lipid and glucose homeostasis, improvement of fibrosis	[58,59]
	FGF21	WAT/BAT, liver, adipose tissue, skeletal muscle,	Decrease of plasma triglycerides levels and sugar intake	[60,61,64]
HPS	HPS	Liver, skeletal muscle	Regulation of cell growth, induction of IR	[65,66,68]
RBP	RBP4	Various peripheral tissues, liver	Regulation of hepatic de novo adipogenesis	[70,71,72]
IGF	IGF-I	Skeletal muscle	Glucose and lipid metabolism	[75]
IGFBP	IGFBP7	Liver, kidney	Regulation of IGF in circulation, inhibitory tumor	[74,75]
SeP	SeP	Liver	Inhibited hepatic glucose production	[80,81,82]

## 6. Risk Factors Associated with NAFLD and AKI

NAFLD is often considered a progressive disease and involves a range of serious complications, such as lipid accumulation, steatosis, inflammation, fibrosis, cirrhosis, and HCC [86]. The progressive course of NAFLD corresponds to a long disease course, resulting in the presence of multiple predisposing factors for other diseases, such as the MetS [87].

The MetS is widely believed to be associated with unhealthy lifestyle habits, such as excessive nutritional intake, inactivity, and resulting obesity [88]. Traditionally, NAFLD was considered to be a result of MetS, because MetS could cause abnormal liver function.. According to the current view, there is a bidirectional relationship between NAFLD and various components of the MetS, especially in T2DM and hypertension [89]. Based on clinical observations, the disease consequences of MetS tend to be more severe in patients with NAFLD, and some studies indicate that most NAFLD patients die from cardiovascular disease and cancer rather than from complications related to the liver itself [90]. 

Many current studies define being overweight and obese based on the body mass index [91]. Obesity and its associated diseases, such as T2DM, CKD, and various cancers, are among the leading causes of death worldwide. Reports from the World Health Organization indicate that excessive fat accumulation in overweight and obese individuals may damage in ways that are not yet known. Obesity is inextricably linked not only to the MetS, but also to NAFLD and a range of cancer-type-related diseases, and together with IR, is a risk factor for the development of HCC [92]. IR is a disease state that is clinically manifested by the patient’s inability to respond to an increase in their glucose uptake, even after stimulation with exogenous or endogenous insulin. The liver is the organ that links NAFLD and IR, and it is generally accepted that the body is more likely to develop IR when excess fat is accumulated in the liver. This close association between NAFLD and IR suggests that one can reverse-predict the risk of developing NAFLD by examining IR [93,94,95].

AKI is a global public health problem. Various risk factors are associated with AKI, such as surgical or traumatic infection, hypovolemic shock due to bleeding, sepsis, drug intake, traumatic surgery, oxidative stress, and abnormal apoptosis [9]. Further, AKI risk factors are not only associated with traditional clinical contexts, such as surgery or use of intravascular iodine-containing contrast agents, but are also closely related to the individual’s genetic makeup, age, sex, potential comorbidities, and immune status [96]. In the last two years, research on these risk factors has added energy metabolism dysfunction as a key element in the pathogenesis of AKI.

## 7. Hepatic Factors in NAFLD

NAFLD alters the hepatic secretion of many proteins, and a major proportion of these proteins affect the insulin response. Further, the liver can interfere with the physiological state of some extrahepatic tissues, such as skeletal muscle and fat, through inflammation, fatty acid oxidation, and other pathways that together contribute to the IR status of the body. The liver can influence insulin secretion by the pancreas not only via the extrahepatic factors described above, but also by independently influencing glucose utilization by peripheral tissues via protein secretion [97].

The liver acts as an endocrine organ that secretes hepatic factors. Because of its unique structure and ability to regulate blood flow, the liver can send molecular signals to and interact with distant tissues via circulation. Dysregulation of inter-organ pathways centered on the liver may lead to misregulation of energy homeostasis and, eventually, to metabolic dysfunction or new diseases. Explaining the mechanisms that regulate the expression of hepatic factors and their linkage to distant tissues or organs in NAFLD is essential for understanding inter-organ interactions and developing therapeutic strategies to treat metabolic dysfunction [98]. NAFLD is accompanied by lipid accumulation in the liver and steatosis, and it is hypothesized that this alters the expression of hepatic factors. As mentioned above, targeted studies have identified several hepatic factors that affect peripheral metabolism, including fetuin-A [45,46,47], ANGPTL6 [39,40,41], FGF21 [66,67], and SeP [83,84,85]. This demonstrates that hepatocytes or hepatic factors are involved in metabolic cross-talk and suggests the possibility that protein signals originating from fatty liver cells communicate with other cells to regulate the metabolic phenotype. Although there are many patients with impaired kidney function induced by liver disease, whether hepatokin plays a role in kidney disease is still unknown.

## 8. AKI in NAFLD

The liver receives approximately a quarter of the cardiac output, which, because of the high level of blood perfusion, allows the redistribution of hepatic factors to other tissues. Because of the unique structure and regulation of blood flow in the liver, proteins secreted by hepatocytes and non-parenchymal cells within the liver concentrate in the hepatic sinusoids, and then circulate to the heart through the central vein to the inferior vena cava, from which they are redistributed to peripheral tissues. NAFLD alters the expression of a variety of hepatic proteins and thus affects peripheral metabolism [97]. AKI in patients with cirrhosis is generally classified into two categories. The first type is hepatorenal syndrome-AKI, i.e., a liver-induced impairment of kidney function during the compensatory phase of liver injury. Deterioration of renal function occurs in the absence of nephrotoxic drugs and intrinsic nephropathy. The second type is AKI without the hepatorenal syndrome, which is not strongly associated with the liver and is mainly caused by inadequate renal perfusion, disturbance in bile acid metabolism, and the use of nephrotoxic drugs during the course of the disease [99]. Because of the proven link between organs, dysfunction of organs other than the kidney may have multiple adverse effects on the kidney, causing a recurrent course of AKI with poor treatment outcomes. Clinical data suggest that major organs, such as the lung, heart, brain, and liver, may potentially engage in cross-talk with the kidney. Therefore, dysfunction of any of these organs may increase the risk of AKI. Recent studies have identified a number of mechanisms by which other organs interact with the kidney during the onset of AKI, which allow for an abnormal activation of the immune cells, massive increase in inflammatory factor levels, mitochondrial disorders, and abnormal apoptosis in the kidney (Figure 2). Conversely, AKI could prevent organs other than the kidney from functioning properly [100].

Studies have shown that various hepatic factors, such as ANGPTL4, ANGPTL6, FGF21 and IGF-1, secreted by the liver are important for alleviating metabolic dysfunction and may have positive implications for the prevention of AKI. These hepatic factors are differentially expressed in the liver with steatosis, which may lead to a disruption of energy homeostasis and, ultimately, to metabolic dysfunction, such as AKI. The current treatment of AKI depends on its cause: renal volume resuscitation is used for pre-renal AKI, intravenous albumin, vasoconstrictors for hepatorenal syndrome AKI, and supportive therapy for acute tubular necrosis [101].

Based on the observations that the expression of hepatic factors changes in disease and that hepatic factors regulate metabolic phenotypes of distal tissues or organs, we propose that analogs of beneficial hepatic factor or inhibitors of deleterious hepatic factors could be developed in the future as a treatment for AKI or other metabolic diseases that occur during NAFLD.

## 9. Advances in Early Diagnostic Biomarker Research for AKI

If the regulation of hepatic factor expression is important for AKI prevention, then the use of early diagnostic biomarkers for AKI is an essential test of therapeutic efficacy. These biomarkers indicate a strong possibility of AKI before it occurs and accurately reflect the progression of renal damage in AKI.

Because of the slow pace of development of breakthrough biomarkers for the early diagnosis of AKI, currently, clinicians still grade AKI based on Scr levels and urine volume. However, the lack of sensitivity of these two to kidney injury and the severe lag between them render them inefficient for diagnosis [9,21]. In 2015, the International Club of Ascites (ICA) launched a revised consensus on the diagnosis and management of AKI in patients with cirrhosis (ICA2015) based on extensive review of the relevant reports in the nephrology field. The consensus innovatively proposed dynamic examination of Scr levels as an early diagnostic index of AKI, addressing the diagnostic deficiency of this marker. However, this did not alleviate the difficulty of diagnostic identification of AKI.

A few markers of kidney injury have been established, such as KIM-1, NGAL, TIMP-2, and IGFBP7. KIM-1 is one of the markers of early kidney injury. This protein can cross the cell membrane. When renal tubular cells are injured by ischemia, the expression of KIM-1 increases, and the protein rapidly enters the tubular lumen from the extracellular fraction to protect the injured kidney [102]. Elevated KIM-1 levels in urine hint at renal tubular injury. Importantly, elevated KIM-1 levels in urine precede any changes in Scr and are associated with the degree of kidney injury and inflammation, similar to KIM-1 levels in the blood. Another maker, NGAL, is normally abundant in the cytoplasmic granules of neutrophils and is not highly expressed in other cells. However, upon tubular injury, NGAL expression increases, inhibiting apoptosis and promoting tubular cell proliferation [103]. Further, TIMP-2 and IGFBP7 are extremely useful for early diagnosis because they are both involved in the repair of renal injury [104]. Importantly, as a biomarker, IGFBP7 exhibits extremely high sensitivity and accuracy for the early response to kidney injury and has a very promising translational potential.

Overall, the above proteins are more highly expressed in patients with AKI than in those without AKI, and their parallelism with the traditional markers Scr and urea nitrogen supports their credibility as early diagnostic biomarkers for AKI. However, some challenges remain for biomarker selection, and the complexity of AKI causation in the clinical setting and differential marker selections lead to the development of different detection methods. A combination of multiple markers may be a good solution for the time being to improve diagnostic accuracy, but more research is needed to address the problem of early diagnosis of AKI.

## 10. Discussion

As the liver and kidney are both extremely important for systemic homeostasis, impairment of the function of one organ can induce pathogenesis affecting the other organ. Newer studies and clinical observations have revealed several biochemical and molecular pathways that may implicate the liver in the development of AKI, including alteration of liver enzyme profiles, oxidative stress, activation of inflammation, and apoptosis. The molecular interactions linking the liver to the kidney that are not organ-based are supported by research findings. Hepatic factors that alleviate the metabolic dysfunction upon hepatic steatosis include ANGPTL4, ANGPTL6, FGF21, IGF-1, etc. These proteins regulate adipocyte activation to increase energy expenditure, increase insulin sensitivity and glucose uptake by the body, reduce plasma triglyceride levels, and regulate cholesterol homeostasis. In other words, they reduce the energy overload of the body to reduce the possibility of induction of other metabolic diseases.

The question of how to maintain systemic homeostasis in the event of NAFLD can be approached in two major ways. We can control the development of metabolic disease by controlling the abnormally elevated levels of liver proteins during the disease, including some ANGPTL family proteins, fetuin-A, heparin, etc. This would control the development of metabolic disease by improving insulin sensitivity and maintaining lipid homeostasis to suppress systemic inflammation and IR. Dysfunction of organs other than the kidney is likely a non-negligible factor in the development of AKI and its poor prognosis, which explains why AKI often occurs in combination with other serious illnesses. Clinical case data suggest that major organs, such as the lung, heart, brain, and liver, are potentially involved in a cross-talk with the kidneys. Hence, their dysfunction may increase the risk of AKI. During the onset of AKI, the interaction of the other organs with the kidney is perturbed, which results in abnormal immune cell activation, massive increase in inflammatory factors, mitochondrial dysfunction, and abnormal apoptosis of a large number of functional kidney cells. Interestingly, these are also potential mechanisms by which AKI affects the function of organs other than the kidney.

We hypothesize that the association between NAFLD-mediated hepatic factors and AKI is mainly two-fold: First, NAFLD, the expression of a variety of hepatic factors that are beneficial to the systemic metabolism is altered, as a result of dysregulation of immunity, inflammation, and oxidative stress, all of which are closely related to the occurrence of AKI. Second, since the liver receives a major portion of the total cardiac blood output, it can distribute hepatic factors throughout the body via the circulation with the possibility of inter-organ interaction to regulate metabolic phenotypes. This suggests a link between hepatic factors and AKI in patients with NAFLD, involving IR, lipotoxicity, oxidative stress, and inflammatory factors, although the specific pathways and mechanisms should be supported by further studies.

## 11. Conclusions

In this review, we first summarized the role of the major hepatic factors, then explored the connection between NAFLD and AKI based on their mechanisms of action, and finally proposed flexible regulation of the expression of circulating hepatic factors perturbed in disease to control metabolic disease as a treatment for NAFLD. In the foreseeable future, circulating hepatic factors as a means to regulate metabolic homeostasis will show great application prospects.

## Figures and Tables

**Figure 1 ijms-23-12464-f001:**
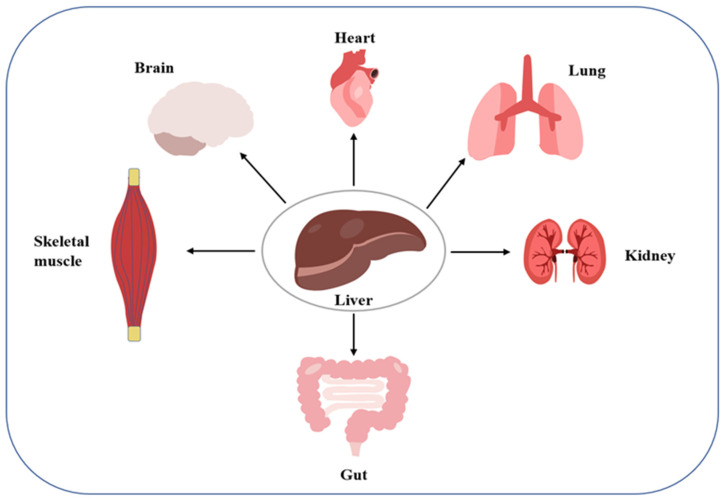
Conceptual map of organ communication, with the liver at the center of communication under physiological conditions. Depicted are the liver-heart, liver-lung, liver-kidney, liver-gut, liver-skeletal muscle, and liver-brain.

**Figure 2 ijms-23-12464-f002:**
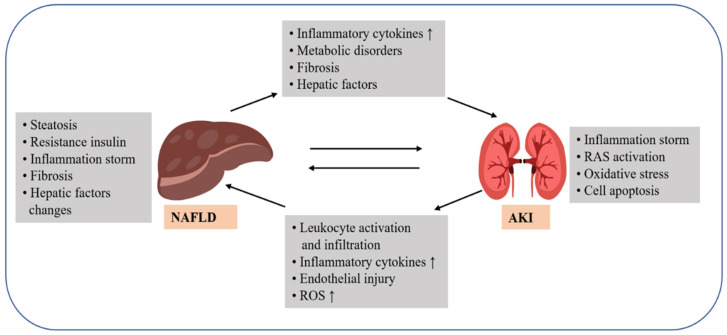
Various mechanisms associated with non-alcoholic fatty liver disease (NAFLD) and acute kidney injury (AKI). Liver-kidney interactions during NAFLD include changes in circulating levels of beneficial hepatic factors and metabolic disorders, which result in renal injury via lipogenesis and inflammatory responses. In turn, the renal response exacerbates oxidative stress and cytokine storm, causing liver damage. RAS, renin-angiotensin system; ROS, reactive oxygen species; ↑: Enhanced effect.

## Data Availability

Not applicable.

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
