# Peer review of "Critical Overview of Hepatic Factors That Link Non-Alcoholic Fatty Liver Disease and Acute Kidney Injury: Physiology and Therapeutic Implications"

_ijms, 2022, doi:10.3390/ijms232012464_

Round 1

Reviewer 1 Report

In the present manuscript the authors review the epidemiology and association between NAFLD and AKI. The topic is interesting as the focus has mostly been on CKD in previous studies. I have the following comments:

-          Please double-check the manuscript as there are many spelling/grammar mistakes and typos (e.g the sentence “Previous studies have revealed that liver-derived factors beare transported via the circulation liver to distal organs, such as the kidney, where they elicit specific effects”

-          Please revise the following sentence in the abstract “the global prevalence of NAFLD increases, with the accompanying high mortality rate”, because NAFLD is not a condition characterized by a high mortality rate

-          When discussing the prevalence of NAFLD, apart from the meta-analysis from Younossi et al from 2016, I would also mention more recent studies (see DOI: 10.1111/liv.14828)

-          One might also mention the association between liver stiffness as a measure of fibrosis in NAFLD and albuminuria and CKD as shown recently (doi: 10.3390/biom12010105 and doi: 10.1016/j.dld.2021.02.010). In general I would stress more the impact of the histological severity of NAFLD on kidney function.

Author Response

Dear Reviewer 1,

代表我的合著者,我们非常感谢您给我们机会修改我们的手稿,我们非常感谢您对我们题为"连接非酒精性脂肪肝疾病和急性肾损伤的肝脏因素的批判性概述:生理学和治疗意义"的手稿的积极和建设性的意见和建议。(ID:国际通用汽车工业协会-1950617)。

感谢您的宝贵建议。论文中的主要更正和对审稿人评论的回应在回复信中。(回复信已作为附件上传)。

Reviewer 2 Report

This review describes the systemic effects of several hepatic factors, with a particular focus on their association with hepatic factors whose expression is altered in NAFLD and AKI. Both the content and structure of the article are well organized, and I think it is a paper that takes a novel approach. However, there are some errors in some words in the text, so please review it again. For example, fetuin-b and fetuin-B are cosxisting in chapter 5.2, title of chapter 7, Hepatic factors under and NAFLD, areIt in L.524, an sugar intake (FGF21), and hepaticde (RBP4) in Table 1, and so on.

Author Response

亲爱的审稿人2,

代表我的合著者,我们非常感谢您给我们机会修改我们的手稿,我们非常感谢您对我们题为“连接非酒精性脂肪肝疾病和急性肾损伤的肝脏因素的批判性概述:生理学和治疗意义”的手稿的积极和建设性的意见和建议。(ID:国际通用汽车工业协会-1950617)。

感谢您的宝贵建议。论文中的主要更正和对审稿人评论的回应在回复信中。

谢谢你和最好的问候。

您真诚的,

通讯作者:臧红梅

邮箱:zanghongmei@ahmu.edu.cn

Round 2

Reviewer 1 Report

No further comments